# Quantitative analysis of visual codewords of a protein distance matrix

**Jure Pražnikar** [1,2]*, **Nuwan Tharanga Attygalle** [1]

**1** Faculty of Mathematics, Natural Sciences and Information Technologies, University of Primorska, Koper, Slovenia, **2** Department of Biochemistry, Molecular and Structural Biology, Institute Jožef Stefan, Ljubljana, Slovenia

* jure.praznikar@upr.si

**Data Availability Statement:** Dataset is available at open-access repository ZENODO. URL: https://zenodo.org/record/5906637 DOI: 10.5281/zenodo.5906637.

## Abstract

3D protein structures can be analyzed using a distance matrix calculated as the pairwise distance between all Cα atoms in the protein model. Although researchers have efficiently used distance matrices to classify proteins and find homologous proteins, much less work has been done on quantitative analysis of distance matrix features. Therefore, the distance matrix was analyzed as gray scale image using KAZE feature extractor algorithm with Bag of Visual Words model. In this study, each protein was represented as a histogram of visual codewords. The analysis showed that a very small number of codewords (~1%) have a high relative frequency (> 0.25) and that the majority of codewords have a relative frequency around 0.05. We have also shown that there is a relationship between the frequency of codewords and the position of the features in a distance matrix. The codewords that are more frequent are located closer to the main diagonal. Less frequent codewords, on the other hand, are located in the corners of the distance matrix, far from the main diagonal. Moreover, the analysis showed a correlation between the number of unique codewords and the 3D repeats in the protein structure. The solenoid and tandem repeats proteins have a significantly lower number of unique codewords than the globular proteins. Finally, the codeword histograms and Support Vector Machine (SVM) classifier were used to classify solenoid and globular proteins. The result showed that the SVM classifier fed with codeword histograms correctly classified 352 out of 354 proteins.

## Introduction

The analysis of protein structures using the distance matrix of Cα atoms has a long history in structural biology. To date, a protein distance matrix has been used for structural alignment, protein classification, and finding homologous proteins [1, 2]. Recently, tremendous progress has been made in predicting 3D proteins based on distance matrix and artificial intelligence [3–6]. Various studies have shown that the representation of protein structure in 2D space has the following main advantages: it represents local, short-, medium-, and long-range contacts between Cα-atoms simultaneously and is rotation and translation invariant [7]. The protein distance matrix contains the distances between residues, which can be represented as a gray-scale image, where the distances between pairs of Cα-atoms are represented by intensity. Therefore, feature extraction can be applied to obtain points of interest.

**Funding:** JP was supported by Javna Agencija za Raziskovalno dejavnost RS, award number P1-0048 and I0-0035-2790. NTA was supported by Javna Agencija za Raziskovalno dejavnost RS, award number I0-0035-2790. The funders had no role in study design, data collection and analysis, decision to publish, or preparation of the manuscript.

**Competing interests:** The authors have declared that no competing interests exist.

In previous studies on distance matrix, the extracted features have been used to classify proteins and explore the universe of protein folding space. For example, the DALI Z-score [1, 8], the Local Feature Profile [9], and the Effective Moment Feature Vector [10], which were generated as three-dimensional maps by dimensionality reduction, have shown that all α-, all β-, α/β-, and α+β-domains are grouped. Moreover, multi-view rendering of 3D protein structures was used to obtain a 2D image followed by feature extraction [11]. In this method, the codebook is generated by clustering visual words and a histogram is used to explore similarities between protein structures. Taken together, these studies have shown that feature vectors or histograms of distance matrix are valuable to explore and investigate the space of protein folding in reduced dimensions. The feature vectors or histograms can be obtained in many different ways, but the common approach in the above literature is the method from computer science—Bag-of-Visual-Words. Sivic and Zisserman first introduced Bag of Features [12] for matching objects in videos, where they investigated whether it is possible to retrieve a video scene that contains a particular object. In addition, Yu *et al* [13] developed an image retrieval system based on the Bag of Features method.

Several studies also introduced image recognition techniques to successfully classify protein structures using Speeded Up Robust Features (SURF) and Scale-invariant Feature Transform (SIFT) descriptors. The agreement between proteins was measured using the similarity of all feature descriptors rather than codeword histograms [14, 15]. This approach does not introduce a fixed codebook size, but determines the smallest distance between all feature pairs from two images. A high number of matching points indicates that two images have similar patches and we can conclude that they were generated from similar 3D protein structures.

Much of the previous research on distance matrix features has focused on optimal vocabulary size, various feature extraction algorithms, protein folding universe representation, and classification performance. Basic patterns in distance matrix showing secondary structure elements (α-helix, parallel and anti-parallel β-sheet) are well known, but less work has been done on quantitative analysis of distance matrix features. For example, Choi *et al* pointed out that the "null" pattern (15th mediodis) is the most common [9]. In the mentioned study, one hundred medoids (submatrix of size 10x10) were extracted from the training set (sampled proteins). Thus, 100 medoids represent a local feature of the protein distance matrix. However, no analysis was performed on the frequency of all 100 mediodis, the most frequent position in the distance matrix (map to image), or the correlation between the number of unique features and the solenoid proteins.

Thus, our main motivation was to extend the analysis of distance matrix features instead of focusing only on the speed and accuracy of classification. Therefore, the focus of the present study is on a quantitative analysis of the codewords of the protein distance matrix, and we tried to answer the following questions: What is the relationship between the frequency of the codeword and its position in the distance matrix, what is the distribution of the codewords, and what is the correlation between tandem repeats proteins and the number of unique codewords. For our study, we used bag-of-visual-words, a method commonly used in image classification, where a histogram of codewords represents each image. The aim of this study is to (i) find the optimal vocabulary size of KAZE algorithm for fast protein classification of SCOPe protein domains, (ii) explore the most frequent position of visual codewords in the distance matrix, (iii) investigate the relationship between unique visual codewords, domain size, solenoid and tandem repeats in proteins.

## Methods

### SCOPe domain database and (non) solenoid proteins

The set was taken from the SCOPe 2.07 database [16–18] and included four main classes: all α, all β, α/β and α+β. We excluded domains that are not a true family and are labelled as X.X.X.0;

we also excluded all individual members at the family level. Thus, each SCOPe family in the analysed database contains at least two members. The dataset contains 7308 distance matrix images, and we extracted 2,929,445 KAZE features. The images of the distance matrix contained the distances between all $C\alpha$-atoms. Therefore, each distance matrix was represented as a grayscale image in the range [0, 1], where 0 (black) represents long-range contacts and 1 (white) represents short-range contacts.

## Solenoid and non-solenoid protein data set—Benchmark

For the classification of globular and solenoid proteins, we used a benchmark database consisting of 105 solenoid and 247 globular proteins downloaded from the following website: http://old.protein.bio.unipd.it/raphael/precompiled.html [19, 20]. The dataset contains 105 solenoid proteins for which the maximum sequence identity is 35% (CATH 'S' -level, sequence families). In addition to the solenoid proteins, the dataset contains 247 globular proteins with different topological or folding families and no detectable sequence similarity (CATH 'T'-level). Thus, the dataset contains non-homologous proteins and has been used for other methods, e.g.: RAPHEL [20], ReUPred [21], ConSole [22], which are briefly introduced below.

RAPHAEL uses geometric data instead of amino acid sequence profiles. The coordinates of the $C\alpha$-atoms are used to build a coordinate profile. To obtain the coordinate profile, two filters are used, where the first window size for averaging is 6 and the second window size is 3. Then, the period is calculated from the coordinate profile, which is defined as the distance between successive local maxima. In the last step, the Support Vector Machines fed with the feature profile is trained to classify proteins.

ReUPred (Repetitive Units Predictor) is a tool for the prediction and classification of repeating units in proteins. The ReUPred algorithm takes as input the Structure Repeat Unit Library (SRUL), derived from RepeatsDB, and the query protein. ReUPred uses an iterative divide-and-evaluate approach to identify structural units by comparing the protein structure to a manually curated library of repeat units. The alignments found by the repeat-protein annotation that satisfy the predefined similarity criteria are considered valid.

ConSole uses the information contained in the protein contact maps. The 3D protein structure is represented as a contact matrix in which there is a contact between two amino acids if the distance between any pair of heavy atoms is less than 4.5Å. The typical feature of the solenoid contact map is the second diagonal line indicating regular short distance contacts between amino acids. The template matching or normalized cross-correlation performed on the contact map is used as input for classification by a Support Vector Machines.

## Bag of visual words

The earliest reference for Bag-of-Words was in Zellig Harris' Distributional Structure [23]. Since then, the bag-of-words model has been used in various document classification methods. The frequency of each word (number of occurrences) is used as a feature for training a classifier. The Bag-of-Features model (Bag-of-Visual-Words) is influenced by the Bag-of-Words model and is used to classify images by treating image features as words. This Bag-of-Visual-Words algorithm includes the following four main steps: (i) feature extraction, (ii) visual dictionary, (iii) codebook generation, and (iv) histogram generation. All four steps are shown in Fig 1 and in the following text. Bag-of-Visual-Words was generated using the Matlab function *BagOfFeatures*. The image database and Matlab code [24] are in Supplement.

**Feature extraction.** Features are parts or patterns of an image that can be used to classify the image. Instead of storing the entire image, it is efficient to store the unique parts of the image. Feature extraction is the process of extracting unique elements of the image. The

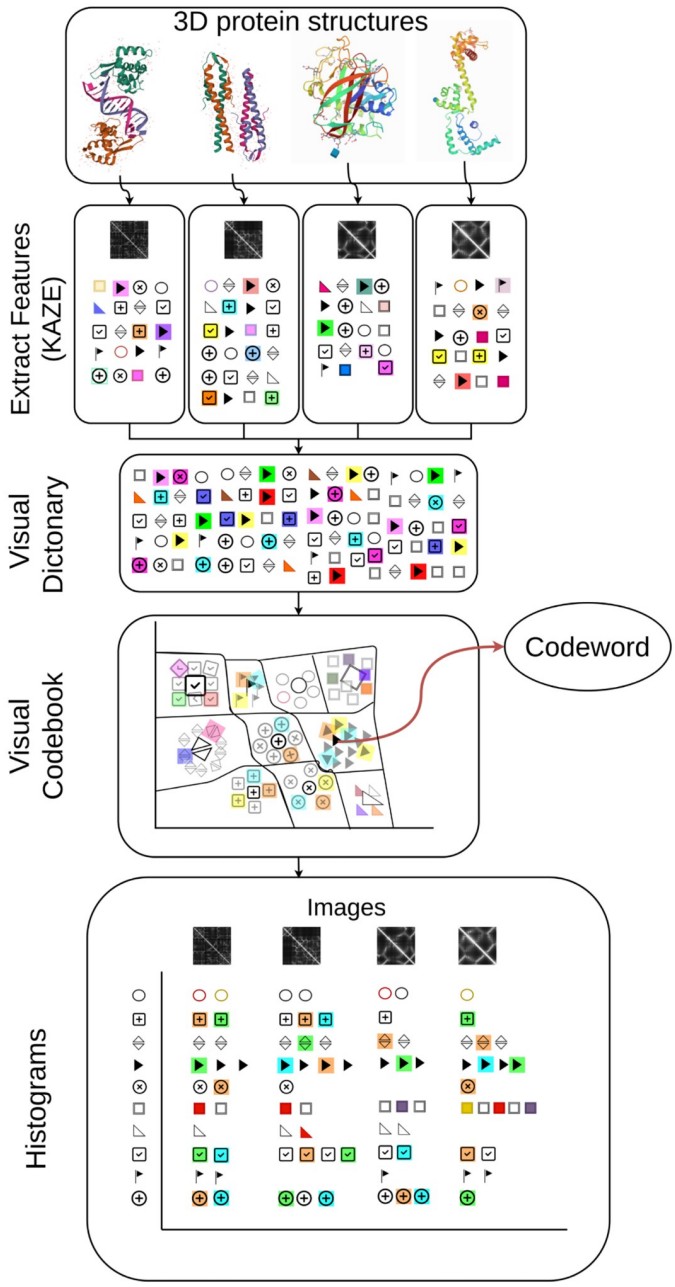

**Fig 1. Graphical representation of Bag of Visual Words.** The presented algorithm consists of four steps: feature extraction, creation of a visual dictionary, creation of a codebook and finally hardcoding or histogram creation.

feature extraction algorithm detects key points and creates descriptors based on the key points. Usually, a descriptor is an array of 64 or 128 elements. We can consider these descriptors as small images extracted from the image. Feature detection algorithms such as Speeded Up Robust Features (SURF), Scale-invariant Feature Transform (SIFT) and KAZE create multiple instances of the same image by blurring and transforming (rotating, scaling, skewing) it. Then the algorithm identifies the keypoints by searching for similar points in all the generated images. Once the algorithm finds the keypoints, the neighbouring pixels around the keypoints

are called descriptors. SURF and SIFT use Gaussian blur to create blurred versions of the images. However, the Gaussian blur smooths both image details, such as object boundaries, and noise to the same extent. Therefore, Gaussian blur degrades the quality of keypoints by reducing localization accuracy and discriminability. In contrast, the KAZE algorithm [25] locally adapts the blur to the image in a nonlinear scale space by using nonlinear diffusion filtering. The KAZE algorithm reduces noise while preserving object boundaries, resulting in better localization accuracy and discriminability. Therefore, KAZE features are more expensive to compute than SURF and SIFT, but outperform SIFT and SURF in terms of feature matching efficiency [26]. Therefore, we used the KAZE algorithm to obtain features from protein distance matrix images.

**Visual dictionary.** In this step, we merged the raw features from all images using the KAZE algorithm. Each extracted feature consisted of a 64-dimensional vector representing the neighbouring pixel values around a given key point. We extracted features $f_i$ from all images in the dataset and stored the extracted features in a collection $F = \{f_1, \ldots, f_M\}$, where $M$ is the total number of features in the entire set. This feature collection $F$ is called a visual dictionary.

**Codebook generation.** The codebook is a compact version of the visual dictionary. Therefore, the codebook stores only a subset of the features $F$ instead of all the extracted features. We grouped similar features of the visual dictionary into clusters using k-means clustering and extracted the centroids of each cluster. Clustering is an unsupervised machine learning approach that groups similar data points. Data points in the same group should have similar features, while data points in different groups should have very different features. The goal of k-means is to cluster similar data points and help users discover the underlying patterns. K-means initialises the centroid locations and then performs iterative computations to optimise the centroid locations for each data point. We grouped the similar features of the visual dictionary into clusters using k-means clustering and extracted the centroids of each cluster. Then we stored the centroids in a new collection, which is called the codebook. Each data point of the codebook is called a codeword.

**Generating a histogram.** Using the codebook, we represented all the images as histograms, i.e., one histogram per image. Each bin of the histogram represents each codeword of the codebook. So, after hard coding, the dimensions of histograms are same for all images. First, we extracted the features from the image and found the closest codeword for each feature in the image (hard coding) and thus created the histogram or feature vector. After creating the histogram, the similarity of the images is calculated as the cosine distance between two histograms—feature vectors. The histogram representation has the same dimension for the entire dataset, regardless of the dimension and number of features of the images.

## Support vector machine classification

The labelled (globular and solenoid proteins) distance matrix images were classified using the *trainImageCategoryClassifier* function in Matlab [24]. The function *trainImageCategoryClassifier* trains a multiclass classifier (Support Vector Machines) using a bag-of-features object. Support Vector Machines (SVM) is a supervised machine learning model that can be used for linear and nonlinear classification problems. SVM is based on the idea of finding a hyperplane in a high-dimensional space that best separates features into different classes (S1 Fig). The features closest to the hyperplane are called support vector points, while the margin width is the distance between the separator and the closest points of each class. The SVM algorithm finds the optimal hyperplane when margin width is maximum. In the present study, an SVM classifier (linear kernel) was used to discriminate solenoid and globular proteins. The input data were histograms of codewords, where each histogram corresponded to a protein distance matrix.

## Results and discussion

### Vocabulary size of a distance matrix

When creating vocabulary, it is crucial that the size of the vocabulary is not too small or too large. If the vocabulary is too small, it will not represent all the features or patches from the image database, so images that are significantly different from each other may have similar codeword histograms. On the other hand, if the vocabulary is too large, quantization artefacts may occur, i.e., overfitting. To test how the classification accuracy depends on the size of the vocabulary, we performed a nearest neighbour classification with different vocabulary sizes.

The cosine similarity measure was used to determine the similarity between two feature vectors—codeword histograms. The minimum cosine distance between two codeword histograms thus defined the nearest neighbour between the query domain and all other domains in the database. The prediction was correct if the nearest neighbour belonged to the same class (or fold or superfamily or family). Fig 2A shows the accuracy as a function of vocabulary size for the class, fold, superfamily, and family levels.

It is obvious that the accuracy is highest for class prediction and lowest for family prediction. The explanation is that the differences between classes are much larger than the differences at the family level of the same superfamily, which makes it harder to predict the correct family level. For example, if the query protein is classified by SCOPe as a.1.1.1 and the closest member was classified as a.1.2.1, the prediction is correct at the class and fold levels, but incorrect at the superfamily and family levels. Moreover, there are only four distinct classes at class level (α alpha, all β, α+β, and α/β), but there are 534, 870, and 1602 distinct classes at fold, superfamily, and family levels, respectively (Table 1). It follows that classification at the fold, superfamily, and family levels is more challenging than at the class level, where there are only four distinct classes.

Further, we can also see that accuracy increases as the vocabulary size increases until a plateau is reached. This is observed at the fold, superfamily and family levels. The most significant increase is observed at the family level, where accuracy increases from 0.72, 0.79 and 0.80 for vocabulary sizes of 1000, 5000 and 10000 respectively. When the vocabulary size increases from 5000 to 10000, the accuracy at the family level increases by only ~1%, and at the class, fold, and superfamily levels, the increase in accuracy is even smaller. It is worth noting that creating and using a larger vocabulary increases the computation time. The creation of bag-of-visual-words with a vocabulary of 1000, 5000 and 10000 codewords required a CPU time (3.4 GHz processor) of about 20, 27 and 39 minutes, respectively. Therefore, for further analysis,

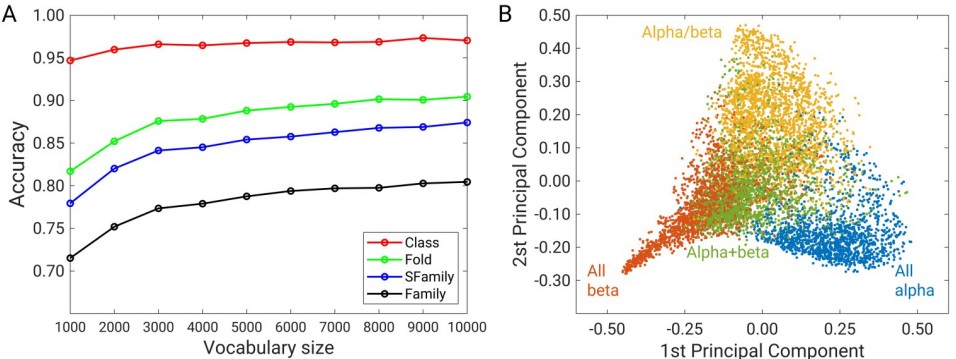

**Fig 2.** (A) Accuracy *versus* vocabulary size of the SCOPe 2.07 database. (B) The first and second principal components of the codebook (7308 SCOPe domains by 5000 codewords).

**Table 1. SCOPe 2.07 levels and the number of unique classes at each level used in this study.**

| Level | Unique classes |
|---|---|
| Class | 4 |
| Fold | 534 |
| Super-family | 870 |
| Family | 1602 |

we set the number of visual words in the codebook to 5000, which gives a good prediction and is not too large to prevent overfitting. Moreover, the principal component analysis of all histograms, i.e., the matrix of size 7308 (proteins) times 5000 (codewords), clearly shows four principal classes (Fig 1B). The first principal component analysis shows low values for all β-domains and high values for all α-domains, whereas α+β and α/β are scattered around the zero value. It follows that the first principal axis discriminates all α- and all β-classes, but not α/β and α+β. The second principal component discriminates between the α+β- and α/β-classes. It is also observed that three groups (all β, all α and α/β) cluster on the positive or negative principal axis, while the α+β class is scattered closer to the origin. We have shown here that the extraction of descriptors in an image of the distance matrix using the KAZE algorithm, followed by the construction of the codebook of 5000 words, gives a good distribution of domains in the low-dimensional space and a good prediction of nearest neighbours, and is consistent with the studies of Hou *et al*, Choi *et al*, and Shi *et al*, who studied the folding space and structural similarity in proteins [8–10].

## Relative frequency and spatial distribution of codewords of a distance matrix

The spatial relationship between codewords is lost in the creation of the codebook because the information about the coordinates of the features is not contained in a bag of visual words. If we know which cluster (codeword) a feature belongs to, we can extract the coordinates of the codewords and relate the coordinates of the codeword with its relative frequency. Here, we examined the relationship between the relative frequency of codewords and the coordinates in the protein distance matrix.

Fig 3A shows the relative frequency of codewords ordered from highest to lowest relative frequency. We see that codewords with the highest relative frequency are present in more than

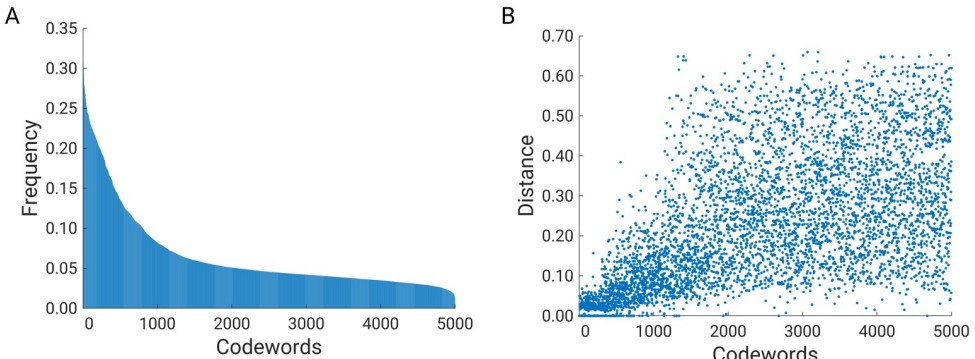

**Fig 3. (A) Barplot of relative frequency of codewords, (B) distance between feature coordinates and diagonal of distance matrix.**

30% of SCOPe domains. Only 52 (~1%) codewords have relative frequency above 0.25, and about 3000 codewords (from 2000 to 5000) are present in less than 5% of the protein distance matrix images. The next question we want to answer is: where are the codewords located in the 2D matrix? Due to the hard coding, each extracted feature belongs to a particular codeword, i.e., a member of a particular cluster, and each feature also contains the x and y coordinates—the centre of the feature. In this way, we can calculate the minimum distance between the feature coordinates and the diagonal of the distance matrix and relate the distance to the relative frequency of a codeword.

Note that the size of the distance matrix depends on the size of the protein, but the shape of the information presented remains the same. The distance matrix always shows local, short, intermediate, and long-range contacts between Cα-atoms. Since protein domains differ by size, we normalized the coordinates to a value between 0 and 1. Note that the maximum distance between the diagonal and the farthest points is limited by the value $\sqrt{2}/2 \approx 0.7$. The zero distance indicates that the feature is on the diagonal, while the ~0.7 distance indicates that the feature is far from the diagonal and represents the farthest contact between Cα atoms.

Fig 3B shows the distance (codeword-diagonal) for all 5000 codewords sorted by relative frequency (from highest to lowest). In Fig 3B, it can be seen that codewords with high relative frequency (from 1 to 500) are closer to the diagonal than codewords with low relative frequency (from 500 to 1000). Less frequent codewords (from 1000 to 5000) are even more scattered and are in the interval from 0.05 to 0.65. To get a better visual impression, we plotted the positions of all features on a normalized distance matrix. The panel plot shows the position of codewords from 1 to 500 (Fig 4A), 501 to 1000 (Fig 4B), 1001 to 2000 (Fig 4C), and from 2001 to 5000 (Fig 4D). It can be clearly seen that the codewords with the highest relative frequency are located on the diagonal and represent local protein structure interactions. The highest density is located in the core of the diagonal, while the lowest density is found at the ends of the diagonal. Fig 4B shows features representing short-range contacts; these are non-diagonal codewords. Nevertheless, there are still some feature points at the tails of the diagonal. The tails of the diagonal contain information about local interactions; however, these codewords are relatively less frequent than the codewords in the core of the diagonal (Fig 4A). The explanation for this is that the codewords at the tails of the diagonal represent flexible terminus. Obviously, more codewords are required for disordered regions than for the rigid core region of the protein domain. In addition, Fig 4C and 4D show the mid- and long-range contacts. In Fig 3D, there are no elements near the diagonal, and the highest density of codewords is in the corners of the distance matrix, which represents the longest contacts between Cα-atom pairs in the protein structure.

## Unique words ratio

This section focuses on how protein domain size is related to the total number of codewords, the number of unique codewords, and the relationship between unique codewords and the repetition of the structural motif within the protein.

The correlation between the number of codewords in the protein and protein domain length was R = 0.96, and the correlation between protein domain length and unique codewords was R = 0.93 (Fig 5A and 5B). It was expected that protein length would correlate positively with the number of (unique) codewords, since a larger image contains a larger number of features. However, when constructing a histogram, different features can be encoded in the same codeword. One reason for this is clustering, where similar but not identical features belong to the same cluster. The second reason is that a protein with repeating units in 3D space also has repeating units in the contact map. Therefore, a large protein with a high

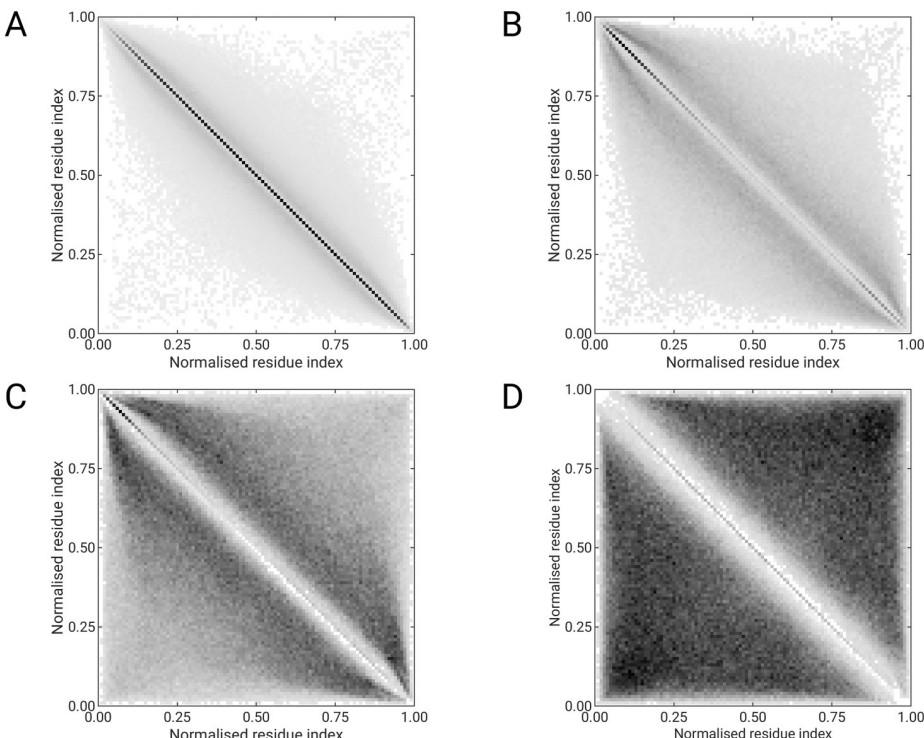

**Fig 4. Distribution of codewords in the distance matrix sorted according to the relative frequency (from highest to lowest).** (A) Visual codewords from 1 to 500, (B) 501 to 1000, (C) 1001 to 2000, (D) and from 2001 to 5000.

number of repeating units may have a low number of unique codewords, but still have a high total number of codewords.

Fig 6A shows the ratio of unique codewords to all codewords for all protein domains in the SCOPe dataset. When all features belong to different codewords (cluster classes), the ratio of unique words is 1, but when a large number of features belong to the same codeword, the ratio of unique words is low. For example, if the distance matrix image contains 500 features extracted by the KAZE algorithm, and these features belong to 400 unique codewords (taken from the precomputed codebook), then to calculate the proportion of unique words you need

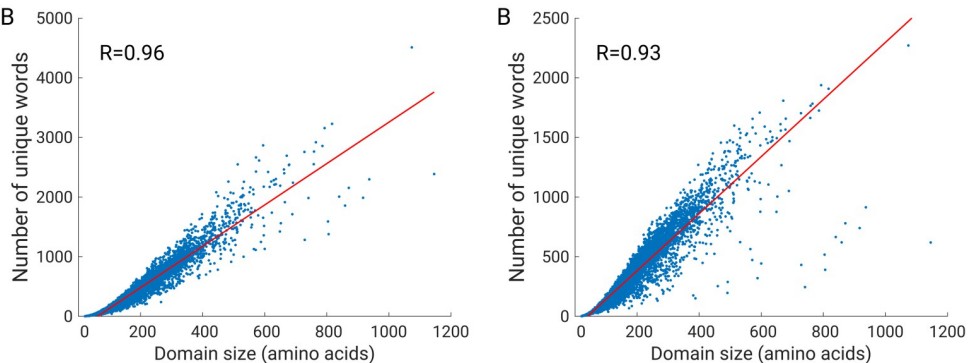

**Fig 5. (A) Number of codewords *versus* domain size, (B) the number of unique codewords *versus* domain size.**

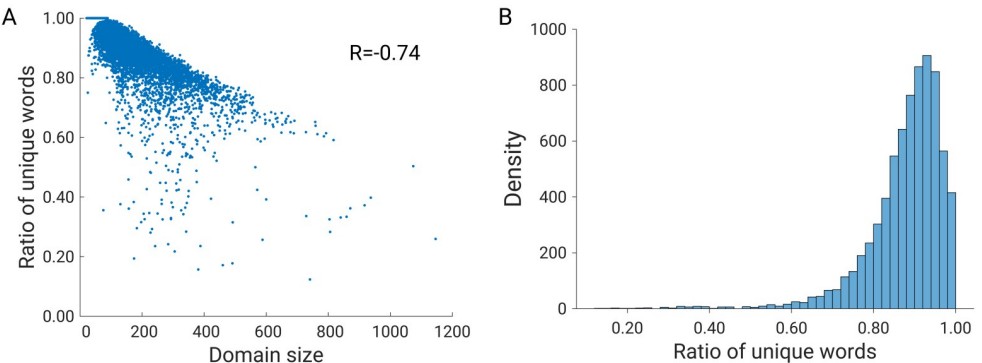

**Fig 6. (A) Ratio of unique codewords *versus* domain size, (B) distribution of the ratio of unique codewords.**

to divide 400 by 500 and get 0.8. The value 0.8 means that 80% of the codewords are unique. We can see that the ratio of unique codewords can be as high as 1.0 and on the other hand it is lower than 0.2. Moreover, we can observe a negative correlation (R = -0.74) between the domain size and the ratio of unique words. The value 0.2 means that only 20% of the features are unique. The histogram (Fig 6B) of the ratio of unique words shows that most (~0.96%) protein domains have a ratio between 0.7 and 1.0, and only a few (~0.6%) domains have a ratio below 0.4, indicating that the protein domain with a low ratio of unique codewords has many repeat units in 3D space. We selected three domains with unique codeword ratio below 0.2 and three domains with unique codeword ratio above 0.92.

As can be seen in Fig 7, the protein domains with a low ratio of unique codewords are solenoid domains containing repetitive structural units (see Fig 7A–7C); in contrast, all three selected domains with a high ratio of unique codewords represent globular proteins (Fig 7D–7F). Solenoid proteins and proteins containing tandem repeats [27] have been and still are the subject of extensive research, as they have been shown to play fundamental roles in many

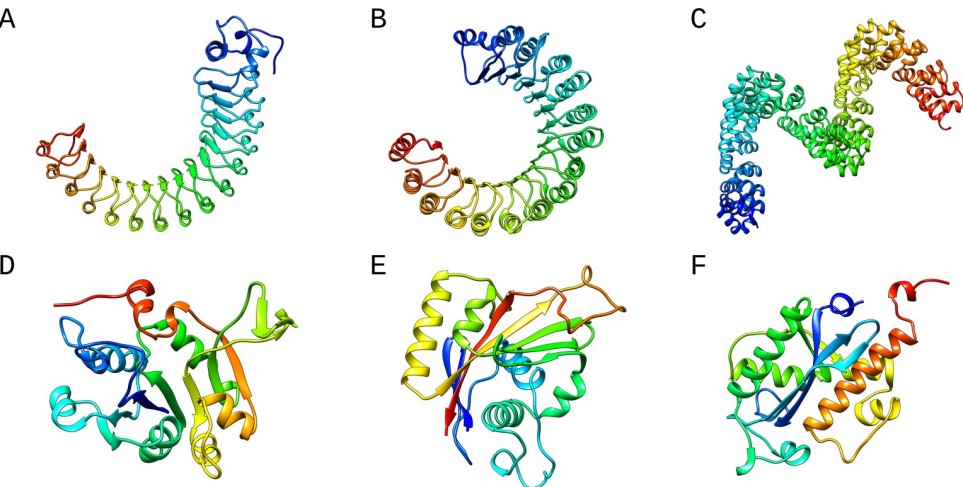

**Fig 7.** (A, B, and C) Domains with a low number of unique codewords (below 0.2), SCOPe domains from left to right: d1o6va2, d2q4gw_, d4cj9a. (D, E, and F) Domains with a high percentage of a unique word (above 0.92) and a domain size above 200 residues, SCOPe domains from left to right: d1w55a1, d2zxea1, d4kb1a_. All ribbons were plotted by UCSF Chimera [33].

biological processes such as signal transduction and molecular recognition [28, 29]. Due to their elongated structures and flexibility, Ankyrin, Leucine-rich, and HEAT -repeat proteins are involved in protein-protein interactions [30]. Fibromodulin, for example, is a Leucine-rich repeat protein of the extracellular matrix that binds to fibrillar collagens and affects fibril velocity [31]. In addition, solenoid proteins also play a crucial role in the interaction with nucleic acids. Exportin-5, for example, is a transporter for microRNA [29]. Therefore, we linked the RepeatsDB (https://repeatsdb.bio.unipd.it/) [32] and our SCOPe dataset. We searched for which domains from our dataset were also included in the RepeatsDB database, which contains annotated tandem repeat protein structures. S2 Fig show that the domains from the RepeatsDB database have a lower ratio of unique words than the domains that are not part of the RepeatsDB database and do not have tandem repeats. In addition, a less strong correlation (abs. value) is observed between domain size and the ratio of unique words in the repeat protein structures (R = -0.63), compared to the domains that are not part of the RepeatsDB database (R = -0.80), see S2 Fig. Overall, these results suggest that the ratio of unique words of solenoid and tandem repeat proteins is shifted towards lower ratios. Moreover, a lower correlation (absolute value) between domain size and unique codewords is observed for solenoid and tandem repeat proteins.

## Discrimination between solenoid and globular proteins

To further investigate the typical ratio of unique codewords for globular and solenoid proteins, we used a solenoid and non-solenoid protein dataset that has been used in many studies and provides a benchmark for distinguishing between globular and solenoid proteins. We used a codebook based on SCOPe domains and calculated the ratio of unique codewords for a benchmark set.

The boxplots in Fig 8 show the ratio of unique codewords for globular and solenoid proteins. We see that globular proteins have a much higher ratio of unique codewords compared to solenoid proteins. This shows that they have a more diverse structure without repeated 3D units. Using the data presented in Fig 8, we see that if the ratio of unique codewords is less than ~0.6, it is likely a solenoid protein.

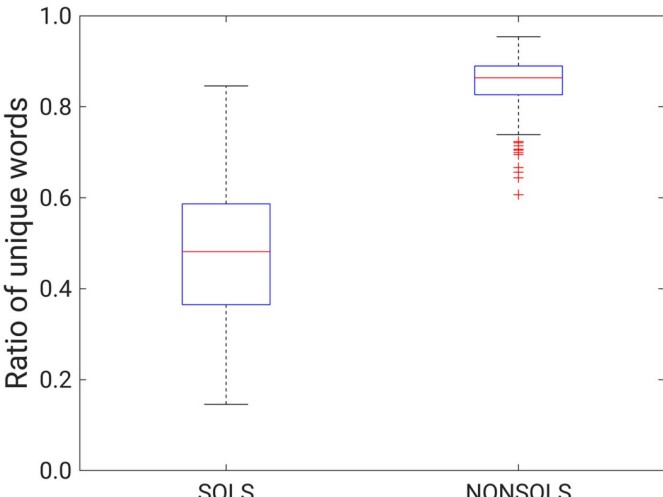

**Fig 8. The ratio of unique codewords of globular and solenoid proteins for a benchmark dataset.**

Furthermore, visual inspection of the histograms of the frequencies of globular and solenoid protein codewords shows clear differences (Fig 9A and 9B). We can see that the histogram of globular protein (Fig 9A) contains many codewords with lower frequencies and there is no significant difference between low and high frequencies. In contrast, the histogram of the solenoid protein contains few codewords with very high frequencies and many codewords with very low frequencies.

To further investigate the performance of bag-of-features classification, we performed an independent classification of globular and solenoid proteins. First, we divided the data into a training and a test dataset, and then created a new codebook (using only the training dataset) in two different sizes: 500 and 1000 words. We randomly divided the dataset into two groups, G1 and G2. The training dataset (test dataset) represented 50% (50%) of the whole dataset. We performed 2-fold cross-validation, i.e., training on G1 and validation on G2, followed by training on G2 and validation on G1. We used the default settings for image classification with the MATLAB function *trainImageCategoryClassifier*, which trains a Support Vector Machine multi-class classifier with a bag-of-features object. The feature vector (codeword frequency histogram, see Fig 9A and 9B for an example) of each protein domain is fed into the Support Vector Machine Multiclass Classifier. The results of the solenoid and non-solenoid protein dataset, which contains 105 solenoid and 247 globular proteins, are shown in Table 2. As can be seen from Table 2, the overall accuracy is over 0.99, which is higher compared to other state-of-the-art methods that achieve an accuracy of 0.96 or less. The Support Vector Machine Multiclass Classifier, which uses a bag-of-features object input, correctly classified 350 out of 352 domains. The largest improvement is observed in the correctly classified solenoids with an increase in accuracy of 12% and 22% compared to RAPHAEL and ReUPred, respectively. Table 2 also shows that this method is very robust to the size of the vocabulary. When the vocabulary was reduced from 1000 to 500 words, the accuracy decreased only slightly, from 99.4 to 99.1.

## Quantitative measure of repeats in protein domain: Inter- and intra-comparison

There are several quantitative measures for comparing protein structures, such as the squared deviation of the coordinates between two superimposed structures, globularity can be estimated by calculating the radius of gyration [34–36] or eccentricity [37], and the locality of contacts between amino acids is measured by relative contact order [38]. In this work, the ratio of

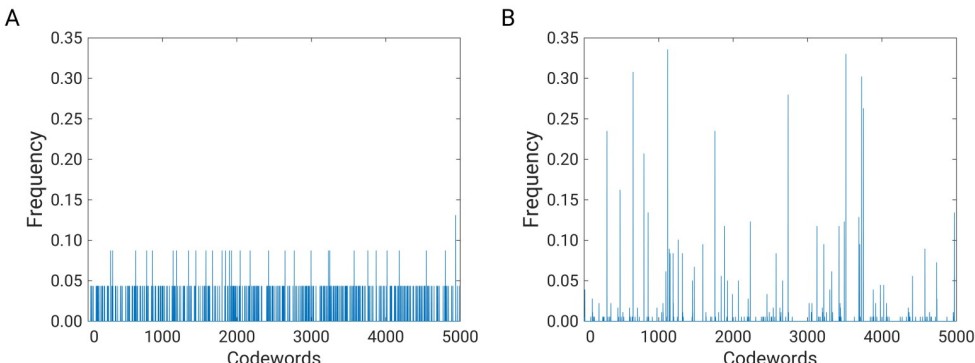

**Fig 9. (A) Histogram of codewords (feature vector) for globular (PDBid:1a79) and (B) solenoid protein (PDBid:1jl5) from the benchmark dataset.**

Table 2. Accuracy of globular and solenoid protein classification using a benchmark dataset.

| | TP | FP | TN | FN | Solenoid | Globular | Accuracy |
|---|---|---|---|---|---|---|---|
| RAPHAEL[a] | 91 | 1 | 246 | 14 | 86.7 | 99.6 | 95.7 |
| ReUPred[b] | 81 | 4 | 243 | 24 | 77.1 | 98.4 | 92.1 |
| ConSole[c] | | | | | 100 | 95 | 96 |
| *This study* | | | | | | | |
| Vocabulary = 1000 | 104 | 1 | 246 | 1 | 99.0 | 99.6 | 99.4 |
| Vocabulary = 500 | 103 | 2 | 246 | 1 | 98.1 | 99.6 | 99.1 |

The data of true positives (TP), false negatives (FP), true negatives (TN) and false negatives (FN) for RAPHEL and ReUPred were taken from their original papers.
[a]RAPHAEL [20].
[b]ReUPred [21].
[c]ConSole [22].

unique words was used as a quantitative measure of regular repeats in protein domains. The ratio of unique words can be used for inter- and intra-comparison of protein domains. Solenoid proteins belong to different SCOPe classes. WD40 and the Kelch motif belong to the all-β class, HEAT and Armadilo repeat belong to the all-α class, and Leucine rich repeat and Ankyrin repeat belong to the α/β and α+β classes, respectively.

Table 3 shows the ratio of unique words for six common protein solenoid domains, where we can see that Leucine rich repeat has the lowest ratio of unique words (0.16), which means that it has the *simplest* and most regular repeats. In this context, simple repetition means that a relatively small number of visual words are required to describe the repetition in the distance matrix. The WD40 and Kelch motifs were expected to have a higher ratio of unique words (0.46 and 0.53, respectively) because the shape is more spherical, resulting in long-range contacts between amino acids that are far apart in the sequence. Consequently, each repeat contains short- and long-range contacts and therefore more visual words are needed to describe a single repeat. Interestingly, the Ankyrin repeat, which does not have a globular shape, has a relatively high proportion of unique words (0.48) compared to Leucine rich motif. Apparently, the repeat consisting of two alpha helices separated by loops, which are also a source of disorder, is more complex than the α/β-horseshoe fold.

In addition to inter-, intra-comparison is also possible to evaluate repetitions in a particular superfamily of the SCOPe database. Fold b.69: 7-bladed beta-propeller contains 15 superfamilies, sorted by date of inclusion in the PDB or publication. We sorted the superfamilies by the ratio of unique words, see S1 Table. As expected, the sorted list of superfamilies is unrelated to the date of entry into the PDB or publication. The comparison between the highest ratio (0.66) and the lowest ratio (0.33) shows a significant difference between the superfamilies. Visual inspection of

Table 3. Unique word ratio for six common protein solenoid domains.

| Solenoid domain | SCOPe id | SCOPe family | Ratio |
|---|---|---|---|
| Leucine-rich repeat | d1a4ya_ | c.10.1.1 | 0.16 |
| HEAT repeat | d1b3ua_ | a.118.1.2 | 0.26 |
| Armadillo repeat | d3bcta_ | a.118.1.1 | 0.35 |
| WD40 | d1pgua1 | b.69.4.1 | 0.46 |
| Ankyrin repeat | d1ixva1 | d.211.1.1 | 0.48 |
| Kelch motif | d1gofa3 | b.69.1.1 | 0.53 |

Domains are sorted by ratio of unique words from lowest to highest.

YVTN repeat (superfamily b.69.2) and Nitrous oxide reductase, N-terminal domain (superfamily b.69.3) reveals that the Nitrous oxide reductase domain has a more complex structure as it also contains alpha helices (see S3 Fig). In addition, the high proportion of unique words may also indicate that some parts of the oligoxyloglucan-reducing end-specific cellobiohydrolase are disordered, indicating a domain irregularity (S3 Fig). As we can see, the ratio of unique words contains information about modularity (extended or globular architecture), complexity, and disorder, and thus quantitatively measures the repeats in the protein domain.

## Conclusion

In this work, we analysed the visual words (codebook) of protein distance matrices. We studied the relationship between the size of the vocabulary and the classification accuracy. It was found that a codebook with several thousand codewords is required for accurate classification. Also, the relationship between the relative frequency of codewords and the coordinates in the distance matrix was analysed. The result was that codewords with higher relative frequency are generally closer to the main diagonal of the distance matrix. We also showed that solenoid domains have a much lower proportion of unique codewords compared to globular proteins, and that the feature vector (codeword histogram) together with a support vector machine classifier can be used very efficiently to discriminate between globular and solenoid proteins. We believe that further work and development can be done to investigate whether the codeword histogram is useful for classifying tandem repeats. In addition, a more advanced approach, such as pooling methods, can be used to incorporate spatial data from protein distance matrix patches.

## Supporting information

**S1 Fig. Support Vector Machines separating hyperplane.** The hyperplane is a function that separates features into multiple classes. The function that separates features in 2D space is a line, while in 3D space it is a plane. When more dimensions are introduced, this function is called a hyperplane. The figure shows two classes of data in 2D space, two support vectors, margin width, and a hyperplane, which in this case is a line.
(TIF)

**S2 Fig.** (A) Ratio of unique words against the number of features. Domains that are members of RepeatsDB and our dataset data set are marked with red dots. (B) The ratios of unique codewords from SCOPe domains (not members of RepeatsDB) and SCOPe domains that are members of RepeatsDB.
(TIF)

**S3 Fig. Three 7-bladed beta-propeller domains (SCOPe fold b.69).** The domains are sorted by the ratio of unique words from lowest (left) to highest (right). (A) YVTN repeat (SCOPe: d1l0qa2) has unique word ratio uwr = 0.33, (B) domain of oligoxyloglucan reducing end-specific cellobiohydrolase (SCOPe: d2ebsa1) has unique word ratio uwr = 0.62 and (C) domain of oligoxyloglucan reducing end-specific cellobiohydrolase (SCOPe: d2iwka1) has unique word ratio uwr = 0.66.
(TIF)

**S1 Table. Unique word ratio for the SCOPe fold class 7-bladed beta-propeller (b.69), which contains 15 superfamilies.** The SCOPe entries were taken from SCOPe 2.07 and are less than 40% identical. The domains are sorted by the ratio of unique words from lowest to highest.
(DOCX)

**S1 Dataset. Distance matrix images.** The supplemental data includes distance matrix images (grayscale, PNG format) and a Matlab script.
(DOC)

## Author Contributions

**Conceptualization:** Jure Pražnikar.

**Data curation:** Jure Pražnikar.

**Formal analysis:** Jure Pražnikar.

**Investigation:** Nuwan Tharanga Attygalle.

**Methodology:** Nuwan Tharanga Attygalle.

**Supervision:** Jure Pražnikar.

**Visualization:** Jure Pražnikar.

**Writing – original draft:** Jure Pražnikar.

**Writing – review & editing:** Jure Pražnikar, Nuwan Tharanga Attygalle.

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
