## [Decision Letter · Decision Letter 0]

25 Nov 2021

PONE-D-21-31856Quantitative analysis of visual codewords of a protein distance matrixPLOS ONE

Dear Dr. Pražnikar,

Thank you for submitting your manuscript to PLOS ONE. After careful consideration, we feel that it has merit but does not fully meet PLOS ONE’s publication criteria as it currently stands. Therefore, we invite you to submit a revised version of the manuscript that addresses the points raised during the review process.

As you will see, both reviewers have issues with the manuscript, both with technical quality and with presentation. One reviewer actually suggested rejection but I would like to give you the opportunity if you think you can address the comments to try to improve the manuscript. The key issues are the scope of the manuscript (would it be better as two manuscripts), the technical issues with the training, and better articulating the implications of the work.

We look forward to receiving your revised manuscript.

Kind regards,

Bostjan Kobe, Ph.D.

Academic Editor

PLOS ONE

“This work was supported by Structural Biology grant P1-0048 and Infrastructure programme grant I0-0035-2790, provided by the Slovenian Research Agency.”

Reviewers' comments:

Reviewer's Responses to Questions

**Comments to the Author**

1. Is the manuscript technically sound, and do the data support the conclusions?

Reviewer #1: No

Reviewer #2: Yes

2. Has the statistical analysis been performed appropriately and rigorously? 

Reviewer #1: N/A

Reviewer #2: Yes

3. Have the authors made all data underlying the findings in their manuscript fully available?

Reviewer #1: Yes

Reviewer #2: Yes

4. Is the manuscript presented in an intelligible fashion and written in standard English?

Reviewer #1: Yes

Reviewer #2: Yes

5. Review Comments to the Author

Reviewer #1: The manuscript submitted by J. Praznikar and A. N. Tharanga describes some features of the protein distant matrix.

Although it is potentially interesting, it is rather confused and a drastic revision is necessary. Actually, this manuscript contains two research projects. The first deals with some features of protein distance matrices and the second is a prediction method of solenoid proteins.

The first part is described superficially and it would be difficult to repeat the calculations for a third part. Methods should be described in more detail, describing the algorithms and not just mentioning them.

The second part is superficial, too. It is unclear if the problem is interesting and useful. I note that the Authors use a database compiled by Tosatto more than a decade ago. This is not acceptable given the PDB dynamics. As a consequence, the results shown in Table 1 might be biased, too.

I strongly suggest the Authors to communicate their results, hopefully in two separate manuscripts, with more details about their methods.

Other issues

Line 112: it is unclear to me what “mediodis” means here.

Line 134: it is unclear why the distance matrices contain the inverse square distances.

Line 214: did the Authors use other similarity measures beside the cosine similarity measure?

Line 247: after “computation time”, it would be interesting to insert some information about the computational requirements (cpu time etc.).

Line 258: “other studies”. The Authors should mention them.

Line 317: perhaps “and solenoid protein domains” might be removed.

Line 336: the “ratio” might be defined by writing an equation.

Line 342: “most” might be associated with a numerical percentage.

Line 343: “few” might be associated with a numerical percentage.

Line 356: “in many biological processes [26-31]” is really very very vague. Might it be possible to expand this discussion.

Line 384: “a database”; which one?

Line 415: “the gold standard database”; which one?

Line 421: The methods “RAPHAEL” and “ReUpred” should be described.

Line 422: “When the vocabulary was reduced from 1000 to 500 words, the accuracy decreased only slightly”; it might be necessary to provide a number.

Reference 14: is there any other publication beside arxiv.org?

Figure 4: titles should be provided to the horizontal and vertical axes.

Figure 5. The expression “(aa)” should follow “Domain size”. Moreover, it might be interesting to insert the slopes of the regression lines.

Minor issues

Line 87: the reference to DALI Z-score is probably wrong. This score has been designed by Chris Sander and Liisa Holm.

Line 96: The reference “Sivic J. et al.” is misspelled (also in the reference list).

Line 130: “[16-18]” should be inserted after “database”.

Reviewer #2: Authors take a refreshingly different approach to protein structure classification by using a (relatively) novel technique of image analysis by "bag of "visual" words" to classify distance maps, a particular visualization approach of protein structures. The paper is well written, results are presented clearly. I have two problems with the paper

- training. No details are given on creation of the training and testing. If, as I assume, the division was done randomly, it is very likely that closely homologous proteins were split between the training the testing set. Such proteins have practically identical structures and hence distance maps. With thousands of features, many of them rare, it is very likely that the system is simply memorizing classifications of such homologs, this is probably easy to test. The training and testing sets must be constructed taking into account the relations between proteins. This is critical to the overall validity of the papers.

- its conclusions are conceptually disappointing. Why introduce a new formalism and do all this work to build a lousy protein structure classifier? The paper provides zero useful information to a structural biologist. What does it tell me that tens of thousands of features are needed to classify a protein. This is a computer science paper written for other computer scientists. If you could appeal to the domain researchers, the paper would be much more valuable

6. PLOS authors have the option to publish the peer review history of their article (what does this mean?). If published, this will include your full peer review and any attached files.

Reviewer #1: No

Reviewer #2: No

---

## [Author Response · Author response to Decision Letter 0]

28 Dec 2021

Dear Editor,

I hereby resubmit a revised and re-edited article. A detailed document containing the authors' response to the reviewers' comments was attached. The changes have been incorporated into the manuscript and are highlighted in yellow. We thank all reviewers for their positive evaluation, reading our paper, and valuable comments to improve the manuscript. Following the reviewers' comments, we have addressed the comments and questions in detail below. 

We have better formulated the implications presented in a new section entitled: “Quantitative measurement of repeats in protein domain: inter and intra comparison". We also stated that the dataset does not contain homologous proteins (sequence identity is less than 35%). We believe that the first part of the paper, which is more related to the development of a new method and the presentation of a new approach to distance matrix analysis, and the second part, which is related to practical examples, are closely related. Moreover, the paper is not too long in our opinion with about 6000 words.

Reviewer #1: 

The manuscript submitted by J. Praznikar and A. N. Tharanga describes some features of the protein distant matrix.

Although it is potentially interesting, it is rather confused and a drastic revision is necessary. Actually, this manuscript contains two research projects. The first deals with some features of protein distance matrices and the second is a prediction method of solenoid proteins.

Methods should be described in more detail, describing the algorithms and not just mentioning them.

ANSWER: We have improved the description of the methods, see updated manuscript. We have added a description of the K-means clustering and Support Vector Machines.

The first part is described superficially and it would be difficult to repeat the calculations for a third part. 

ANSWER: For third parties who have Matlab skills, it would be quite easy to repeat the calculations. The supplemental data includes all 7308 distance matrix images (grayscale, PNG format) and a Matlab script that calculates the bag of features, histograms (feature vector), and classification accuracy. To repeat the calculations for a vocabulary of 1000 words, third parties can invoke the script (BOW_accuarcy.m) by mouse click in Matlab (or run it from the command window). If the user wants to repeat the calculation with a vocabulary of 5000 words, only one parameter needs to be changed ('vocabulary',5000). We also provided a dataset of globular and solenoid proteins (247+105 images, grayscale, PNG format) and a Matlab script (SVM_2fold.m) to classify proteins using SVM. In summary, the supplementary material contains all the inputs needed to run Matlab scripts.

The second part is superficial, too. It is unclear if the problem is interesting and useful. I note that the Authors use a database compiled by Tosatto more than a decade ago. This is not acceptable given the PDB dynamics. As a consequence, the results shown in Table 1 might be biased, too.

ANSWER: Indeed, we used data first compiled by Marsella et al. (2009) and available on the website of Tossato's research group. We used this dataset intentionally because it is a gold standard for distinguishing between solenoid and globular proteins and has been used by many others: RAPHEL, ReUPred, ConSole, Repetita, Wavelet, ...

Another possibility would be to use a new dataset. In this case, we would have to apply all the other methods and algorithms to the new dataset in order to make a comparison between the different methods/algorithms. Anyway, classification was not the only focus of this work. Rather, we introduced an alternative approach to analyse the distance matrix and introduce a qualitative measure of protein repeats.

I strongly suggest the Authors to communicate their results, hopefully in two separate manuscripts, with more details about their methods.

ANSWER: We have improved the description of the methods. The first part of the paper, which is more related to the development of a new method and the presentation of a new approach to distance matrix analysis, and the second part, which is related to practical examples, are closely related. Moreover, the paper is not too long in our opinion with about 6000 words.

Line 112: it is unclear to me what “mediodis” means here.

ANSWER: Medoids are representative objects of a dataset or a cluster within a dataset. The idea of K-medoids clustering is to consider the final centres as actual data points rather than mean data points (K-means clustering). In the mentioned study, one hundred medoids (submatrix of size 10x10) were extracted from the training set sampled proteins). Thus, 100 medoids represent a local feature of the protein distance matrix. In calculating the distance matrix, the authors used a threshold of 20 Å. Larger distances were set to "null", so that a submatrix of size 10x10 contained only "null" elements. To make it clearer for the readers, we rephrase the text as follows:

For example, Soug-Hou Kim et al. pointed out that the "null" pattern (15th mediodis) is the most common. In the mentioned study, one hundred medoids (submatrix of size 10x10) were extracted from the training set (sampled proteins). Thus, 100 medoids represent a local feature of the protein distance matrix. However, no analysis was performed on the frequency of all 100 mediodis, the most frequent position in the distance matrix (map to image), or the correlation between the number of unique features and the solenoid proteins.

Line 134: it is unclear why the distance matrices contain the inverse square distances.

ANSWER: This was a (typo) error. We have corrected it as follows:

The images of the distance matrix contained the distances between all Cα-atoms. Therefore, each distance matrix was represented as a grayscale image in the range [0, 1], where 0 (black) represents long-range contacts and 1 (white) represents short-range contacts.

Line 214: did the Authors use other similarity measures beside the cosine similarity measure?

ANSWER: No, we used default similarity metric (cosine) of Matlab function “retrieveImages”.

Line 247: after “computation time”, it would be interesting to insert some information about the computational requirements (cpu time etc.).

ANSWER: We have added the following sentence in the manuscript. 

The creation of bag-of-visual-words with a vocabulary of 1000, 5000 and 10000 codewords required a CPU time (3.4 GHz processor) of about 20, 27 and 39 minutes, respectively.

Line 258: “other studies”. The Authors should mention them.

Fixed, we mention them.

ANSWER: Fixed, we mention them.

We have shown here that the extraction of descriptors in an image of the distance matrix using the KAZE algorithm, followed by the construction of the codebook of 5000 words, gives a good distribution of domains in the low-dimensional space and a good prediction of nearest neighbours, and is consistent with the studies of Hou et al, Choi et al, and Shi et al, who studied the folding space and structural similarity in proteins.

Line 317: perhaps “and solenoid protein domains” might be removed.

ANSWER: Fixed.

Line 336: the “ratio” might be defined by writing an equation.

ANSWER: We have added an explanation in the manuscript.

For example, if the distance matrix image contains 500 features extracted by the KAZE algorithm, and these features belong to 400 unique codewords (taken from the precomputed codebook), then to calculate the proportion of unique words you need to divide 400 by 500 and get 0.8. The value 0.8 means that 80% of the codewords are unique.

Line 342: “most” might be associated with a numerical percentage.

ANSWER: We add a numerical percentage (~96%), see updated manuscript. 

Line 343: “few” might be associated with a numerical percentage.

ANSWER: We add a numerical percentage (~0.6%), see updated manuscript. 

Line 356: “in many biological processes [26-31]” is really very very vague. Might it be possible to expand this discussion.

ANSWER: We improved text as follows:

Solenoid proteins and proteins containing tandem repeats [25] have been and still are the subject of extensive research, as they have been shown to play fundamental roles in many biological processes such as signal transduction and molecular recognition (Andrade, Marcotte). Due to their elongated structures and flexibility, Ankyrin, Leucine-rich, and HEAT -repeat proteins are involved in protein-protein interactions (Kobe). Fibromodulin, for example, is a leucine-rich repeat protein of the extracellular matrix that binds to fibrillar collagens and affects fibril velocity (Svensson). In addition, solenoid proteins also play a crucial role in the interaction with nucleic acids. Exportin-5, for example, is a transporter for microRNA (Fournier).

Line 384: “a database”; which one?

ANSWER: In Methods, we described the dataset used in previous studies, which serves as the gold standard for distinguishing globular from solenoid proteins. We have expanded the description in the Methods section. We have also added a new section in Methods entitled: "Solenoid and non-solenoid protein dataset - benchmark".

Line 415: “the gold standard database”; which one?

ANSWER: In Methods, we described the dataset used in previous studies, which serves as the gold standard for distinguishing globular from solenoid proteins. We have expanded the description in the Methods section. We have also added a new section in Methods entitled: "Solenoid and non-solenoid protein dataset - benchmark". We also improved the sentence, as follows:

The results of the solenoid and non-solenoid protein dataset, which contains 105 solenoid proteins and 247 globular proteins, are shown in Table 2. 

Line 421: The methods “RAPHAEL” and “ReUpred” should be described.

ANSWER: Done, see section in Methods entitled: "Solenoid and non-solenoid protein dataset - benchmark"

Line 422: “When the vocabulary was reduced from 1000 to 500 words, the accuracy decreased only slightly”; it might be necessary to provide a number.

ANSWER: Done.

When the vocabulary was reduced from 1000 to 500 words, the accuracy decreased only slightly, from 99.4 to 99.1.

Reference 14: is there any other publication beside arxiv.org?

ANSWER: To our knowledge, there is no other publication. 

Figure 4: titles should be provided to the horizontal and vertical axes.

ANSWER: Done.

Figure 5. The expression “(aa)” should follow “Domain size”. Moreover, it might be interesting to insert the slopes of the regression lines.

ANSWER: We added the term "amino acids" to the axis, and also added the regression line.

Line 87: the reference to DALI Z-score is probably wrong. This score has been designed by Chris Sander and Liisa Holm.

ANSWER: The reference of DALI and Z-score (Sander and Holm, 1993 and 2020) is at the very beginning of the introduction. Hou et al. used the DALI Z-score to construct a similarity matrix of 498 protein folds and create a global representation of the protein folding space. 

We have included the reference of Sander and Holm, 1993 in line 87.

Line 96: The reference “Sivic J. et al.” is misspelled (also in the reference list).

ANSWER: Fixed.

Line 130: “[16-18]” should be inserted after “database”.

ANSWER: Fixed. 

Reviewer #2: 

Authors take a refreshingly different approach to protein structure classification by using a (relatively) novel technique of image analysis by "bag of "visual" words" to classify distance maps, a particular visualization approach of protein structures. The paper is well written, results are presented clearly. I have two problems with the paper

- training. No details are given on creation of the training and testing. If, as I assume, the division was done randomly, it is very likely that closely homologous proteins were split between the training the testing set. Such proteins have practically identical structures and hence distance maps. With thousands of features, many of them rare, it is very likely that the system is simply memorizing classifications of such homologs, this is probably easy to test. The training and testing sets must be constructed taking into account the relations between proteins. This is critical to the overall validity of the papers.

ANSWER: The dataset we used is a benchmark dataset that has been used by many other research groups. The dataset contains globular (247) and solenoid (105) proteins where the maximum sequence identity is 35%. Thus, the dataset does not contain homologous proteins. 

We have expanded the description in the Methods section. We have also added a new section in Methods entitled: "Solenoid and non-solenoid protein dataset - benchmark".

- its conclusions are conceptually disappointing. Why introduce a new formalism and do all this work to build a lousy protein structure classifier? The paper provides zero useful information to a structural biologist. What does it tell me that tens of thousands of features are needed to classify a protein. This is a computer science paper written for other computer scientists. If you could appeal to the domain researchers, the paper would be much more valuable

ANSWER: The first part of the paper focused on the general representation and showed that a general classification can also be done using the histogram of codewords (all alpha, all beta, alpha/beta and alpha+beta, see Figure 1). Then we introduced a new qualitative measure - the ratio of unique words. This parameter clearly shows that it tends to be low for solenoid proteins and higher for globular proteins, thus quantitatively describing the repeats in the protein domain. Moreover, the histogram of codewords combined with SVM showed an improvement in discriminating between globular solenoid proteins. And we believe this is valuable for domain researchers. 

However, we feel that the use of the ratio of unique words - a quantitative measure of repeats in the protein domain - is not adequately represented in the study. Therefore, we have added a section titled "Quantitative measure of repeats in protein domain: inter and intra comparison" and included two tables and a figure. Please read and view the improved manuscript.

---

## [Editor Report · Decision Letter 1]

24 Jan 2022

Quantitative analysis of visual codewords of a protein distance matrix

PONE-D-21-31856R1

Dear Dr. Pražnikar,

We’re pleased to inform you that your manuscript has been judged scientifically suitable for publication and will be formally accepted for publication once it meets all outstanding technical requirements.

Kind regards,

Bostjan Kobe, Ph.D.

Academic Editor

PLOS ONE
---

## [Editor Report · Acceptance letter]

27 Jan 2022

PONE-D-21-31856R1 

Quantitative analysis of visual codewords of a protein distance matrix 

Dear Dr. Pražnikar:

I'm pleased to inform you that your manuscript has been deemed suitable for publication in PLOS ONE. Congratulations! Your manuscript is now with our production department. 

Kind regards, 

on behalf of

Professor Bostjan Kobe 

Academic Editor

PLOS ONE